

# Event couple spectral ratio Q method for earthquake clusters: application to North-West Bohemia

Marius Kriegerowski[1], Simone Cesca[2], Matthias Ohrnberger[1], Torsten Dahm[1,2], and Frank Krüger[1]

[1]University of Potsdam, Institute of Earth and Environmental Science, Karl-Liebknecht-Str. 24-25, 14476 Potsdam-Golm
[2]Helmholtz Centre Potsdam German Research Centre for Geosciences - GFZ, Telegrafenberg, 14473 Potsdam

**Correspondence:** Marius Kriegerowski (marius.kriegerowski@uni-potsdam.de)

**Abstract.** We develop an amplitude spectral ratio method for event couples from clustered earthquakes to estimate seismic wave attenuation ($Q^{-1}$) in the source volume. The method allows to study attenuation within the source region of earthquake swarms or aftershocks at depth, independent of wave path and attenuation between source region and surface station. We exploit the high frequency slope of phase spectra using multitaper spectral estimates. The method is tested using simulated full
wavefield seismograms affected by recorded noise and finite source rupture. The synthetic tests verify the approach and show that solutions are independent of focal mechanisms, but also show that seismic noise may broaden the scatter of results. We apply the event couple spectral ratio method to North-West Bohemia, Czech Republic, a region characterized by the persistent occurrence of earthquake swarms in a confined source region at mid-crustal depth. Our method indicates a strong anomaly of high attenuation in the source region of the swarm with an averaged attenuation factor of $Q_p < 100$. The application to $S$ phases
fails due to scattered $P$ phase energy interfering with $S$ phases. The $Q_p$ anomaly supports the common hypothesis of highly fractured and fluid saturated rocks in the source region of the swarms in North-West Bohemia. However, high temperatures in a small volume around the swarms cannot be excluded to explain our observations.

# 1 Introduction

The intrinsic and scattering attenuation of the amplitudes of seismic waves is described by the dimensionless factor $Q$. The mapping of spatio-temporal changes of $Q$ is an important step in seismology, since $Q$ is controlled by temperature, rock porosity, fluid saturation and rock composition (Toksöz et al., 1981). Hence, this factor may help to unravel the possible causes of fluid-induced earthquakes, or thermal anomalies in crustal regions affected by magmatic intrusions. For instance, North-West Bohemia is regularly affected by earthquake swarms lasting several days or weeks with thousands of recorded events with
largest magnitudes up to Ml 4.4 (Fischer et al., 2014). The causes of the repeated earthquake swarms which occur in narrow focal zones remain under debate. Relative earthquake localizations are very precise because of the high waveform quality recorded with a dense local permanent network (Bouchaala et al., 2013). Different tomography studies revealed consistent fig-




ures of the 3D velocity structures (Alexandrakis et al., 2014). The attenuation structure in the source region of the earthquake swarms is scarcely discussed. Some previous studies on whole raypath $Q$ exist and can be used for verification and benchmarking. However, the main aim of this study is to test whether the here developed method can enhance the resolution of near source $Q$ and therefore enable more robust conclusions on source dynamics and the role of fluids in the swarm cycle.

Several studies investigated the regional attenuation of North-West Bohemia by integrating along the full path from sources to receivers. Gaebler et al. (2015) estimated intrinsic and scattering attenuation of $S$ waves ($Q_s$) by means of 14 selected events. Their frequency dependent results indicate mean $\bar{Q}_s$ of approximately 1000. A study by Michálek and Fischer (2013) investigated source parameters and inferred a station dependent, regional $Q_p$ from $P$ phase spectra. They estimated mean $Q_p$ ranging between 100 and 450. They also discuss effects of directivity on $Q$ concluding that the directivity has little influence due to the

position of stations with respect to radiation patterns.

A tomographic study of North-West Bohemia done by Mousavi et al. (2017) indicated a regional average attenuation of approximately $Q_p \approx 100$ to $Q_p \approx 300$ and a pronounced highly attenuative source region where $Q_p < 100$.

Bachura and Fischer (2016) employed two different methods to resolve the regional coda $Q_c$ from the source volume to receivers. They used 13 selected events of the 2011 swarm and found a variation of $Q_c$ between 100 and 2500 within the exploited

frequency range of 1-18 Hz.

A recent work by Wcisło et al. (2018) used a newly developed differential attenuation estimation technique focused on the source region. The authors employed the peak frequency method which relates the half-period of $P$ pulses to attenuation. They also used a differential approach to map the inter-event attenuation using a single station (*NKC*) and found $Q_p \approx 120$ and $Q_p \approx 80$ in the source region.

Most previous $Q$ studies focusing on NW Bohemia were inherently of low spatial resolution. Firstly, either because $Q$ was estimated for the integral ray path between sources and stations (except for the work by Wcisło et al. (2018)) or secondly, because they focused on low frequencies, or both. E.g. Mousavi et al. (2017) used frequencies between 1 to 30 Hz and Gaebler et al. (2015) up to 32 Hz.

In this study, we aim to increase the spatial resolution and to resolve $Q$ for waves traveling only within the small source region

of the earthquake swarms. The developed event couple spectral ratio method is based on the assumption of an exponentially decreasing spectral slope at high frequencies $\omega$ above the corner frequency of the earthquake, often referred to as the $\omega^2$ model (Aki, 1980). From the ratio of the spectral slopes of two events one can estimate the attenuation of P and S phases for the ray path between the two events, given the differential travel time of both events.

The spatially compact seismic clusters in NW Bohemia provide us with a favorable case study scenario due to the high sim-

ilarity of source characteristics (Michálek and Fischer, 2013). We test our method on data recorded from October 6 until October 13, 2008 and a double-difference relocated event catalog of 3841 events with local magnitudes between -0.9 and 3.5 (Fischer and Michálek, 2008). The high density of events during earthquake swarms clustering within a small and confined region allows to infer the local attenuation from event couples, by applying the spectral ratio method (Aki, 1980) to their high frequency amplitude spectra as we will explain in the following section. One major issue when calculating spectral content of

very few data samples is spectral leakage, as a result of the finiteness of the time window under study. In order overcome this



problem Thomson (1982) proposed the multitaper method, which we employ using *mtspec* (Prieto et al., 2009; Krischer, 2016).

## 2 Method

A velocity spectrum $A(\omega)$ of a direct body wave phase originating from a source $j$ recorded at a station $i$ can be described as
(Sanders, 1993):

$$A_{i,j}(\omega) = S_j(\omega)I_i(\omega)R_i(\omega)G_{i,j} \cdot e^{\frac{-\omega t^*}{2}} \qquad (1)$$

where $\omega$ is the angular frequency. $S_j(\omega)$ describes the source spectrum and $I_i(\omega)$ the instrument response. $R_i(\omega)$ is the receiver site effect. $G_{i,j}$ is the frequency independent geometric loss. The exponential term depends on the angular frequency $\omega$ and $t^*$, the path integrated attenuation from the source to the receiver:

$$t^* = \int Q^{-1}/v \, ds \qquad (2)$$

with $Q$ as the dimensionless quality factor, velocity $v$ of the medium and $ds$ a segment along the ray path from the source to the receiver. Attenuation is considered here as a combination of intrinsic and scattering losses. Instead of estimating a total $t^*$ describing the full ray path's attenuation we estimate a local $t^*$ from velocity spectra of two earthquakes sharing the greater part of their ray paths from the seismogenic zone to a receiver. Site effects as well as the receivers response functions cancel
out when two spectra recorded at the same site are analyzed by means of amplitude ratios. Let $A_0$ and $A_1$ be two velocity amplitude spectra of events $E_0$ and $E_1$ (in the following referred to as *first* and *second* event of a couple) recorded at a station $j$ (Figure 1). Assuming that their source spectra $S_0(\omega)$ and $S_1(\omega)$ are similar, taking the natural logarithm of the spectral ratio of $A_{i,0}$ and $A_{i,1}$ yields:

$$ln(A_{i,0}(\omega)/A_{i,1}(\omega)) = \frac{G_{i,0}}{G_{i,1}} - \omega t^*/2 \qquad (3)$$

This equation describes a linear relation with frequency independent geometrical losses to the left of the negative sign. The slope $k$ of a line fitted to equation 3 can be used to derive the attenuation time $t^*$ in between the two sources from which $Q^{-1}$ can easily be inferred using equation 2.

The far field amplitude spectrum $A(\omega)$ of $P$ and $S$ phases can be parameterized as follows: a seismic moment dependent low frequency plateau, the corner frequency $f_c$ and the high-frequency spectral decay approximately proportional to $\omega^2$ resulting
from finiteness of particle rise time and the rupture duration (Aki and Richards, 2002). To remove the dependence on seismic moment we investigate the high-frequency spectral decay, only. Furthermore, exploiting spectral content above the corner frequency also reduces source directivity effects on attenuation estimates (Cormier, 1982).

We infer the corner frequency based on previous studies on NW Bohemia seismic swarms. We use a relation proposed by Michálek and Fischer (2013) to calculate source radii $r$ based on the moment $M_0$ of an event:

$$r = 0.155 \cdot M_0^{0.206} \qquad (4)$$





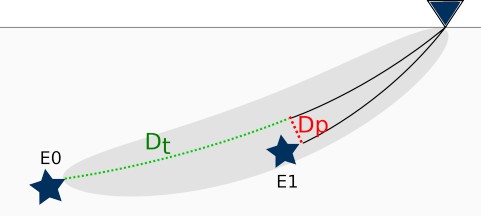

**Figure 1.** Schematic illustration of the geometrical constraints and the employed parameters. The triangle represents a recording station at the surface. Attenuation is estimated for the traversing distance ray path segment ($D_t$, Green dashed line). Geometrical constraints respect the passing distance ($D_p$). Grey shaded area illustrates the Fresnel volume of the first event.

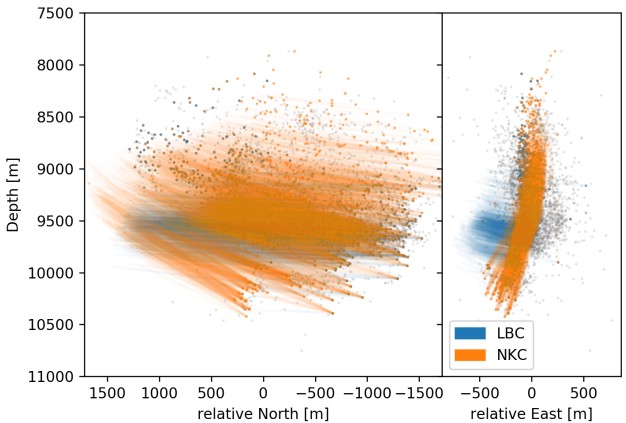

**Figure 2.** Ray segments. Left: Plane view of source slab. Right: side view. The colors indicate the station at which the displayed segment arrived. Grey points show hypocenters which occurred during the investigated time interval but where not attributed to an event couple.

where $M_0 = 1.38 M_L + 10.3$. The resulting source radii $r$ are then converted to $f_c$ using

$$f_c = k\beta/r \tag{5}$$

with $k = 0.32$ (Madariaga, 1976), and $\beta = 3.5$ km/s, which is a reasonable assumption for the source region (Michálek and Fischer, 2013). We increase the lower frequency limit ($f_{min}$) used for our spectral analysis by additional 5 Hz with respect

5  to $f_c$ to account for uncertainties in $M_0$ and to ensure linearity of the high frequency decay. This approach allows frequency bands being wide enough for employing a stable linear regression. The upper frequency limit $f_{max}$ was chosen dependent on the Fresnel volume of a couple's first event (Figure 1) and the upper corner frequency of the anti-alias filter of the recording equipment which is approximately 85 Hz. We calculate the power spectral density using the multitaper spectral analysis method (*MTM*) (Thomson, 1982; Park et al., 1987). With this method the time series is multiplied with several orthogonal slepian

10  tapers which are resistant to spectral leakage. The power spectral density is then reconstructed after Fourier transformation of the tapered samples and a weighted summation of the resulting spectra. A more exhaustive explanation can be found in Park et al. (1987). The applied code is a *Python* wrapper to the *Fortran* routine *MTSpec* (Prieto et al., 2009; Krischer, 2016). *MTM*



achieves stable spectral estimates also for very short time windows. A critical parameter of the *MTM* is the number of slepian tapers as it balances the smoothness and precision of spectral estimates. We use the implementend default, which is

$$N_{tapers} = int(bw \cdot 2) - 1 \tag{6}$$

where $bw$ is the *bandwidth_factor* which we set to $bw = 4$. Lower values prove to increase the number of outliers due to in-
creased spectral leakage. Higher values did not change the results significantly but are more expensive to compute.

We impose strong geometrical constraints to select event couples with respect to a station as sketched in Figure 1. Ray tracing is done based on a 1D velocity model suggested by Alexandrakis et al. (2014) for the seismogenic region combined with a regional crustal model proposed by Málek et al. (2000) (Fig. 4, left panel). The first geometrical constraint is the traversing distance ($D_t$, red dashed line, Fig 1) between an event $E_0$ with respect to perpendicular projections of other hypocenters onto
that path. The second constraint is the passing distance ($D_p$, green dashed line in Fig 1) of that projection of $E_1$ onto the ray between $E_0$ and the station. We defined a minimum traversing distance of $D_t \geq 750$ m to ensure that the signal of the second event is attenuation sufficiently to be detectable in the couple's spectral ratio.

Subsequent to geometrical preselection upper frequency limits of the analyzed bandwidth are potentially corrected to lower values dependent on the Fresnel volume in between event $E_0$ and a station. This guarantees that $E_1$ is located within the Fresnel
volume for the entire analyzed frequency band (grey shaded area in Figure 1).

With this approach we get an estimate for the attenuation along the traversing distance (green dashed line in Figure 1) and when repeated for a large number of event couples can retrieve a median attenuation for the entire source region. Furthermore, we expect rupture dynamics causing random perturbations of the high frequency source spectra to average out. The described method is advantageous over other methods which require handcrafted features like onset duration picking as it can
be automatized given that an onset catalog is at hand.

## 3  Synthetic Study

In order to evaluate the expected number of exploitable couples given the geometrical constraints we calculate the relative number of pairs at discrete surface points covering the region of North-West Bohemia. The size of blue points in Figure 3 represents relative number of pairs based on ray tracing through a 1D layered model (Fig. 4). Largest numbers of pairs are
expected along a North-South striking patch which follows the striking direction of the main fault plane. However, in this case study we use only those stations which provide continuous recordings for the investigated time span. These are stations *KRC*, *LBC*, *NKC*, *SKC* and *VAC*. After geometrical filtering we expect stations *NKC* and *LBC* to produce the highest number of couples since they provide continuous recordings for the entire time period and are in a favorable lateral location. Most other stations are located where no or a negligible number of event pairs are expected. Figure 2 shows the rays which penetrate the
source volume and fulfill the geometrical requirements described above. It shows that for events recorded at the most significant stations *NKC* and *LBC* the highest ray density and therefore sensitivity is in the lower half of the seismogenic zone. This bias is more pronounced for recordings at station *LBC*. Also, these ray segments sample the volume up to approximately 500 m to the West of the seismic swarm.



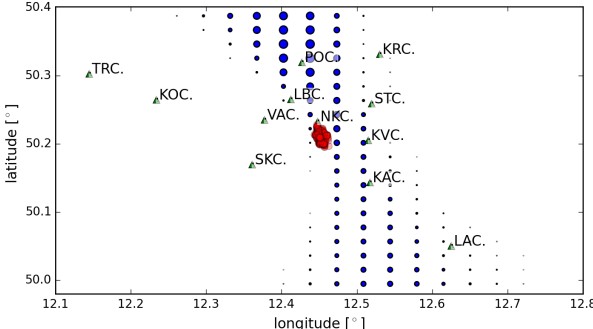

**Figure 3.** Relative number of event couples indicated by size of blue points. Red dots show the seismicity of the investigated swarm. Green triangles indicate locations of the WEBNET stations.

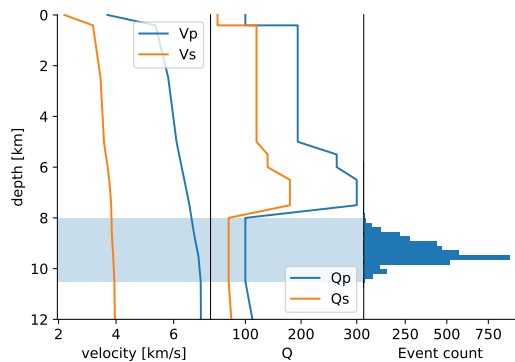

**Figure 4.** Synthetic velocity and attenuation model used for validation of the method. The seismogenic zone is marked by light blue. The attenuation in that zone is expected to be decreased with respect to the regional attenuation model.

We use synthetic waveforms calculated using reflectivity method (Wang, 1999) employing the same 1D velocity model as for ray tracing (Figure 4). Hypocentral locations and origin times are taken from the double difference relocated catalog of Fischer and Michálek (2008). All synthetic sources are double couple sources with mean strike, dip and rake set to $170 \pm 10$, $80 \pm 10$ and $-30 \pm 10$ degrees, respectively, uniformly distributed in all three domains. The mean strike, dip and rake values are the predominant source types stated by Fischer et al. (2014) which were retrieved based on polarity analysis of $P$ phases. The seismogenic zone (depth 8500 m - 10500 m in Figure 4) has a $Q_p$ of 100 and $Q_s$ of 50. It is overlain by a more complex attenuation structure, characterized by higher $Q$ values.

In order to mimic uncertainties in origin times, locations and velocity model travel times are perturbed by 10 ms, uniformly distributed. The uncertainties of the velocity model have an effect only in the source region since both rays of a couple traverse through the same overlaying velocity model.

The window length was 0.15 s for $P$ and 0.3 s for $S$ phases. The minimum allowed cross correlation of event couples in



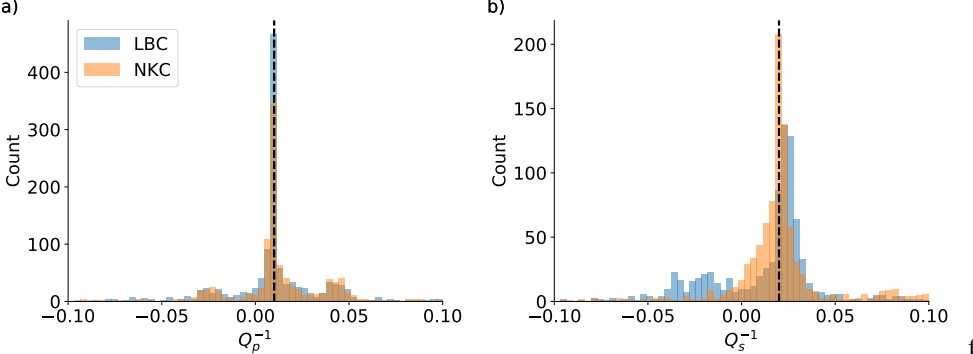

**Figure 5.** Synthetic tests targeting $Q_p = 100$ (a)) and $Q_s = 50$ (b)) with noise free data. The correct values for $Q_p$ and $Q_s$ are indicated by the vertical dashed lines. The station color coding as given in the legend is used consistently throughout all following images.

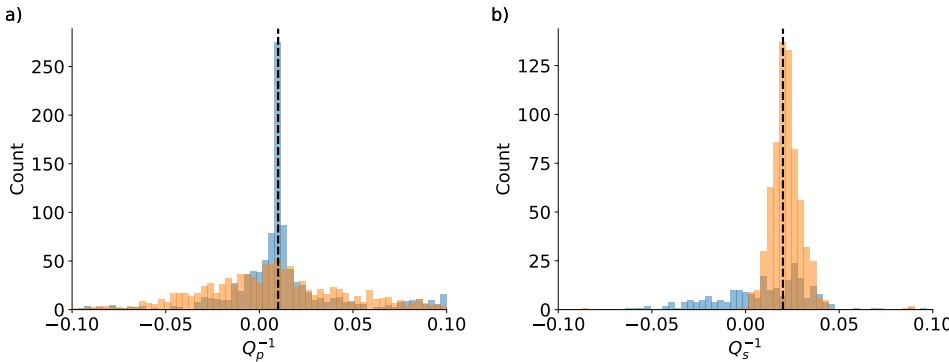

**Figure 6.** Synthetic tests with the same setup as in 5 but with additive real recorded noise from analyzed stations.

synthetic tests and later application to real data was set to 0.75. Signal-to-noise ratio (SNR) of a phase under consideration compared against a noise sample preceding the $P$ phase has to be larger than 5 across the entire selected frequency band (after slight spectral smoothing to reduce effects of spectral notches). These three requirements efficiently reject outliers. The minimum allowed bandwidth is 10 Hz, which excludes all events with magnitudes of less than 0.5, given the magnitude

5    dependent lower frequency limit ($f_{min} = f_c + 5.0Hz$, where $f_c$ is calculated using equation 5). The bandwidth threshold stabilizes the linear fit to the spectral ratio as it limits the minimum number of data points. We evaluate $Q_p$ from vertical channels and $Q_s$ on North-South and East-West components and average results for each couple.

Data availability of the recorded dataset has been accounted for. Synthetic traces were only produced for an event if the recorded dataset contains data as well. All synthetic traces where convolved with the transfer functions of the WEBNET

10    stations to generate realistic velocity traces.

Figure 5 shows distributions of retrieved $Q^{-1}$ estimates from all event couples of the synthetic test using noise-free traces.





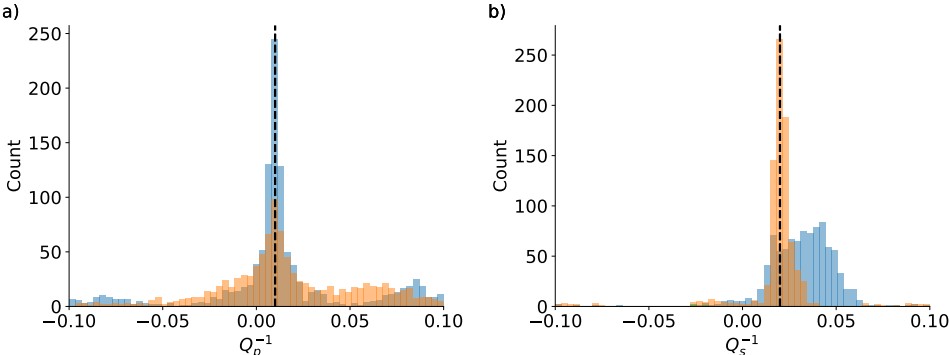

**Figure 7.** Synthetic tests with the same setup as in 6. Synthetic traces convolved with synthetic source time functions.

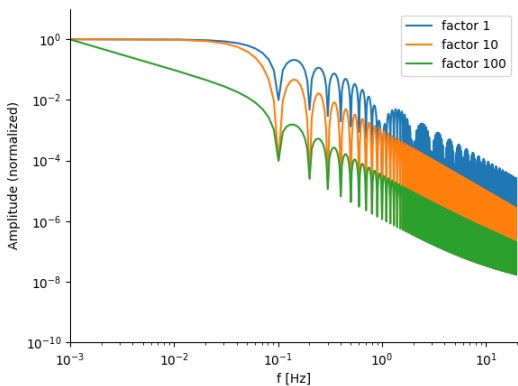

**Figure 8.** Normalized synthetic amplitude spectra of applied source time functions with different durations as a factor relative to the blue one (factor=1).

The distributions show some scattering solutions. A peak in both cases ($Q_p$ and $Q_s$) resemble the targeted attenuation model (grey, vertical line).

The next test depicted in Fig. 6 includes additive recorded noise. Data windows without seismic events in the recorded data have been manually extracted and randomly added to synthetic traces to mimic realistic noise conditions. $P$ phase results show

5    a broadening of the distributions at all stations. While the distribution of station *LBC* still centers around the model value, the results of station *NKC* show only weak correlation with the correct model. This is a result of the location of station *NKC* close to the nodal plane of the dominant rupturing plane where smallest signal amplitudes are expected. $S$ phase results match the model at *NKC* but show strong scattering at *LBC*.

In a next step (Fig. 7) we convolve synthetic Greens functions with realistic magnitude dependent source time functions. The

10   applied source time function is half sine shaped where the slope of the high frequency spectral roll off is not dependent on





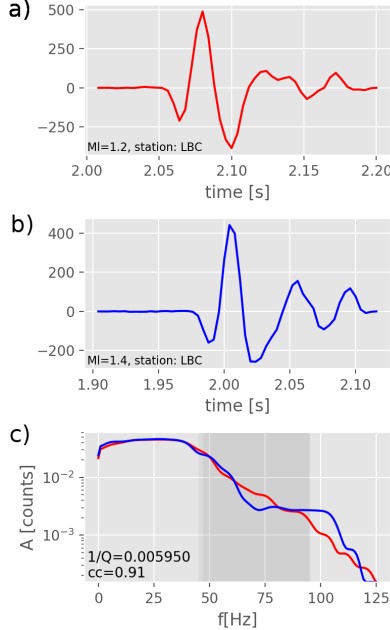

**Figure 9.** Two $P$ phase wavelets recorded at station *LBC* and their spectra. First two panels show a) first and b) second event of the analyzed event couple. The grey shaded area in c) indicates the used frequency band. The cross-correlation coefficient is 0.91 and the attenuation $(Q^{-1})$ was in this case estimated as approximately 0.006

the width of the applied pulse as can be seen in Figure 8 where normalized synthetic source spectra are depicted for different relative pulse widths. The vertical position of the spectral envelope changes with changing duration but the slope remains the same for all depicted factorized pulse widths. Other than expected, this stabilizes results. This is a result of the pulse broadening which leads to a stabilization of *MTM* estimates as onsets become less transient.

5   The performed synthetic tests cannot reproduce waveforms in its full natural complexity. Nevertheless, they prove that the concept is capable to estimate attenuation of the anticipated region.

## 4   Application to North West Bohemia

North-West Bohemia is a favorable case for testing our approach. Several focal mechanism studies on earthquake swarms in this region indicate dominant principle faults striking at 169° and 304° (Vavryčuk, 2011), which have been active in different

10   seismic sequences. Events occurring during a swarm tend to rupture on the same fault. This observation in combination with compactness of seismic clusters (Figure 10) explains the high similarity of waveforms observed for each swarm. We therefore also assume that source characteristics including rupture directivity effects are similar throughout each swarm cycle.

By the time of the 2008 swarm the WEBNET stations were equipped with three component short period seismometers, except for station *NKC* located in the epicentral area which is a broad band station. All waveforms are sampled at 250 Hz. A manual




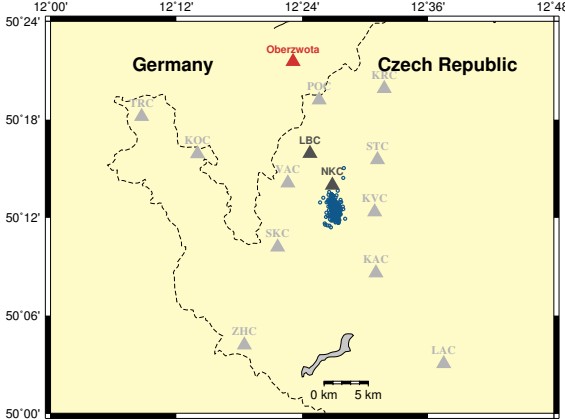

**Figure 10.** WEBNET stations (triangles) and seismicity (blue circles) which occurred in North-West Bohemia during the investigated time period. Red triangle: temporary station installed by University of Potsdam in October, 2017. Stations in Nový Kostel (*NKC*) and Luby (*LBC*) are highlighted. All other stations produce no or insignificant number of data points.

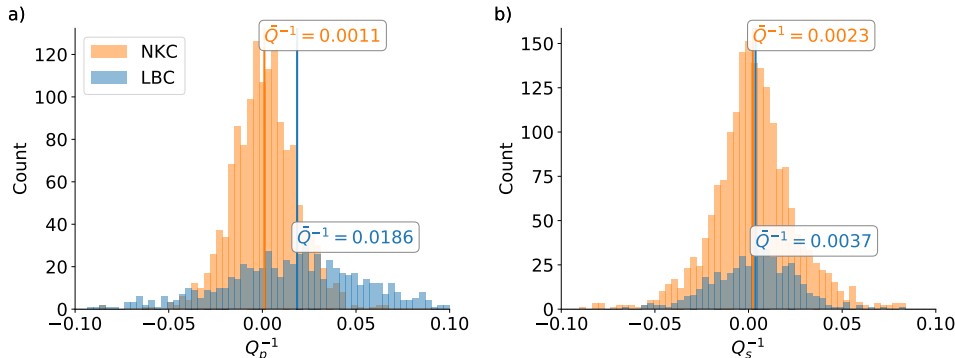

**Figure 11.** Attenuation results at stations *LBC* and *NKC* for $Q_p$ (a)) and $Q_s$ (b)). Median values are indicated by the overlined $Q$.

revision of all event waveforms has been done to remove those which show indications of event doublets happening shortly after each other but not being indicated as such in the catalog. Spurious signal leftovers of a preceding event not necessarily cause high distortion of the fundamental $P$ phase waveform and may thus not be removed by setting a cross correlation threshold. However, their effect lead to distortion at high frequencies of phase spectra and significantly increase the number

5    of outliers during the analysis. The catalog of $HypoDD$ (Waldhauser and Ellsworth, 2000) relocated events is comprised of 3841 events and their associated $P$ and $S$ phase picks. When applied to station *LBC*, a total of 641 couples where used which fulfill the requirements in terms of SNR, cross-correlation and geometrical constraints. Results of $P$ phases evaluated at station *LBC* (Fig. 11, left) have a median $\bar{Q}_p^{-1} = 0.019$, equivalent to $\bar{Q}_p = 53$. The distribution shows some negative results which do not have a physical meaning and are related to noise in spectral estimates. Results retrieved based on data from station *NKC*

10    are significantly more unstable, as Figure 11 (right) indicates. The distribution shows a large number of negative results. The



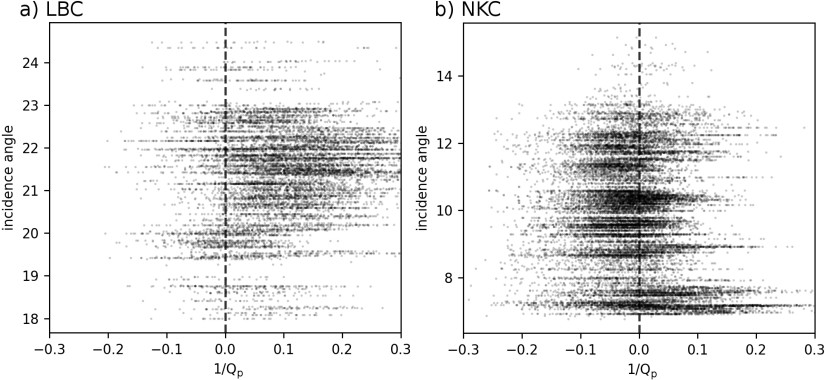

**Figure 12.** Incidence angles plotted against $Q_p^{-1}$ of rays originating from event couples recorded at station *LBC* (left) and *NKC* (right). $Q_p^{-1} = 0$ is highlighted with a dashed line.

median attenuation is $\bar{Q}_p^{-1} = 0.001$, equivalent to $\bar{Q}_p = 1000$. 1404 couples where used in this case.

Attenuation evaluated for $S$ phases show almost zero centered distributions at both stations *NKC* and *LBC* which in turn means significant number of negative and therefore unphysical measures. Median attenuation values are $\bar{Q}_s^{-1} = 0.0023$ ($\bar{Q}_s = 435$) at station *NKC* and $Q_s^{-1} = 0.0037$ ($\bar{Q}_s = 270$) at stations *LBC*, respectively. Both values are comparably large compared to $Q_p$

estimates from station *LBC*. A bias of these $S$ phase attenuation measures is introduced by the $P$ phase coda energy interfering with $S$ phases and therefore distorting the anticipated high frequency content.

 In order to achieve a better understanding of the method's breakdown for $P$ phase recordings of station *NKC* we disable the cross correlation threshold and scrutinize $Q^{-1}$ against a multitude of parameters for station *NKC* and *LBC*. Figure 12 shows incidence angles of rays of event couples on the y- and $Q^{-1}$ on the x-axis. By definition the incidence angle is almost identical

for both events of a couple. It becomes evident that larger incidence angles ($> 8$ degrees) show a tendency to produce negative $Q^{-1}$ while results from events with steep incidence angles produce positive $Q^{-1}$ values. When compared to the same kind of plots for station *LBC* no such trend is evident.

 Station *NKC* is located at the northern edge of the swarm's epicentral region. Hence, incidence angle approximately correlates with latitude, indicating a location dependent problem. When plotting latitude of both events of a couple and color coding $Q^{-1}$

(Figure 13) the results for station *LBC* show, in accordance with Figure 11 mostly positive results and few negative outliers. Results of station *NKC* show a transition from positive $Q^{-1}$ values in the North to negative ones in the South, separated by a gap of event pairs between $lat = 50.211$ and $lat = 50.212$. This trend is mostly dependent on first events (x-axis) of each couple. This implies a systematic change in frequency content from two separated segments of the swarm occurring along raypaths from the source region to station *NKC*.

Figure 14 shows $P$ phase waveforms of first events of couples at station *NKC* and *LBC*, filtered between 1 and 30 Hz for northern (Fig. 14, left panels) and southern (Fig. 14, right panels) events, separated at $lat = 50.2115$. While the used filter frequencies are actually below the exploited frequency band used in the analysis they highlight that the waveform complexity





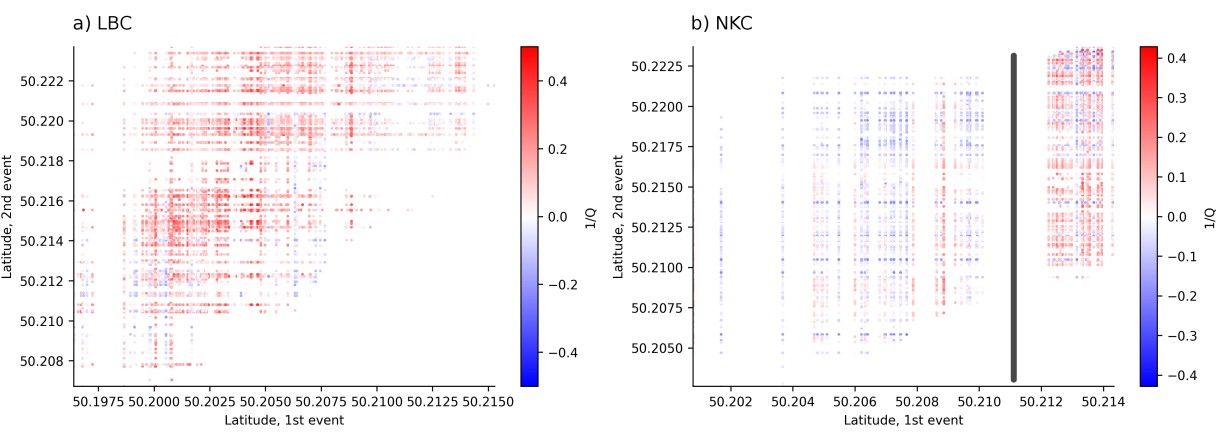

**Figure 13.** Latitudes of both events of each couple on x- and y-axis and color coded attenuation. a) *LBC*, b) *NKC*. A clear separation of negative (blue) from positive (red) attenuation at a latitude of 50.211 degrees becomes evident (vertical bar) with reference to the 1st event (x-axis). Attenuation at station *LBC* is mostly positive.

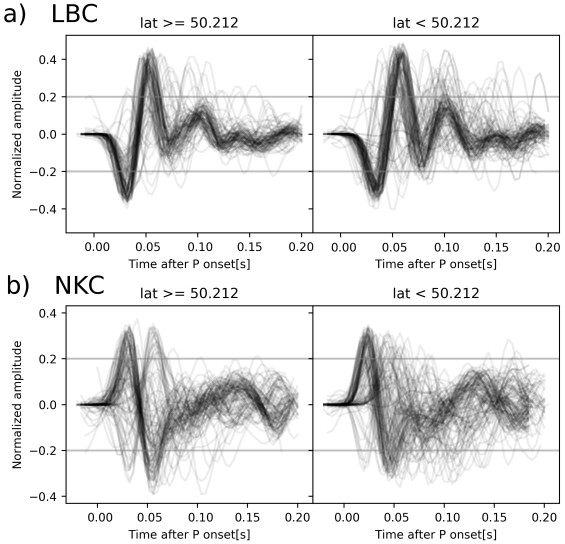

**Figure 14.** $P$ phase onsets of couples' first events in the northern ($lat > 50.212$) and the southern part of the swarm area. a) show recordings at station *LBC*, b) recordings at station *NKC*.

is significantly higher for events at station *NKC* than for those at station *LBC* indicating that scattering plays a major role along the ray paths to station *NKC*. Furthermore, there are $P$ phases with flipped polarities which indicates that the station was situated close to the nodal plane. This aspect is in accordance with synthetic tests in chapter III which show worse performance of station *NKC* then station *LBC* for $P$ phase measures.



## 5   Interpretation and Discussion

We present a newly developed method to estimate attenuation from spectral ratios of event couples. The short analyzed time windows are prone to spectral leakage which we mitigate by applying the multitaper approach. However, this method can only suppress leakage caused by windowing effects. Leakage from source effects such as finiteness of the slip and the rupture are

not smoothed as they are expected source characteristics and not filter artifacts.

The given geometry and available data limit the perceptive field of the applied method mostly to the lower half of the seismic swarm (Fig. 2).

Synthetic tests show that the method is capable to reproduce average source volume attenuation of $Q_p = 100$ and $Q_s = 50$ given the 2008 earthquake swarm hypocentral locations. In noise free condition a precise result can be achieved for both, $P$

and $S$ phases at both significant stations *NKC* and *LBC*. With additive recorded noise, the distribution of $Q_P$ results broadens but still resembles the true attenuation with high precision at station *LBC*. Synthetic waveforms at station *NKC* suffer from weaker *SNR* at high frequencies indicating that the applied *SNR*-threshold of 5 is too optimistic. However, reducing this value would also reduce the number of data points and therefore have a negative impact on the statistical significance of results. $Q_s$ results at station *NKC* are more robust then at station *LBC*. Both measures, $Q_s$ and $Q_p$, improve when convolving synthetic

waveforms with synthetic source time functions as this stabilizes the multitaper spectral estimates.

The application to recorded $P$ phase onsets shows fewer negative $Q_p$ results at station *LBC* and than at station *NKC* which results in a distribution with a clearer offset with respect to $Q_p = 0$. Waveforms recorded at station *LBC* are significantly higher correlated than those at station *NKC* where changing waveform polarities and high waveform complexity can be observed. We hypothesize that rays arriving at *NKC* experienced relatively stronger scattering or that a nearby reflector creates multiples

which interfere with signals recorded at *NKC*. In the latter case, the reflector would have to be situated in a location where interactions with signals arriving at *LBC* are weaker. Mousavi et al. (2017) assume a highly fractured medium in combination with accumulated free gas or fluids. Our findings support this hypothesis. A 3D $V_p/V_s$ tomography by Alexandrakis et al. (2014) identified a low $V_P/V_S$ ratio body directly overlaying the focal zone. The increased waveform complexity seen at station *NKC* can be a result of waveform interaction with that body. The effects do not necessarily have an influence on results

at station *LBC* where rays have different paths and take off angles.

Another influence may be rooted in the different families of focal mechanisms. Vavryčuk et al. (2013) reported three different families of focal mechanisms for the 2008 swarm. While the slope of high frequency spectra should not directly be affected by the radiation pattern, there can be higher order effects like rupture propagation and rupture complexity. Dependent on the take off angle these rupture dynamics can affect the high frequency spectral roll off and therefore map into attenuation estimates

(Kaneko and Shearer, 2015). $P$ phase polarity changes at station *NKC* indicate that the station is located close to the nodal plane of the main rupturing fault. This circumstance can increase the effect of the aforementioned effects seen at station *NKC*. If they differ systematically in the lower and upper source region, this can lead to biases in attenuation analysis due to the heterogeneous sensitivity across the fault plane. Still, we do not see such effects at station *LBC* and therefore speculate, that the dominating effect is the differing raypaths or a combination of both, raypath scattering and rupture dynamics.



Previous studies by Michálek and Fischer (2013) investigated source characteristics in NW Bohemia and suggested station dependent whole path integrated mean $Q_P$ values ranging between 100 and 450. We find lower values which can be a result of hydration of the seismogenic zone. Haberland and Rietbrock (2001) also report highly increased attenuation ($Q < 100$) within earthquake cluster regions and postulated that this could be related to hydration or partial melting. For instance, melt migration

has been postulated from the size and migration pattern of earthquakes of the 2000 earthquake swarm (Dahm et al., 2009). On the other hand, Alexandrakis et al. (2014) interpret their results on velocity variations by dehydration processes. Our results deduced from station *LBC* for average attenuation are in line with previous findings pointing to high attenuation in the source volume.

Frequency bandwidth is a critical parameter which is limited mostly by the corner frequency of the recording setup and signal

to noise ratio at high frequencies. Future plans of the Intercontinental Drilling Project (ICDP) include the installation of up to 4 borehole seismometers in NW Bohemia. It can be expected that our method will benefit from these measurements. Improved signal to noise ratios allow to sample and exploit information at higher frequencies which will stabilize the spectral estimate. Furthermore, higher sampling rates allow a better temporal (and therefore spectral) resolution of $P$ and $S$ phases. This will, in turn, also allow to use even shorter time windows. For the method discussed here, it would be favorable if at least one of

these borehole stations will be situated in a location where a high number of ray path sharing couples can be found. The most sensitive region follows the NNE - SSW striking of the fault and concentrates in the North of the earthquake swarm (Fig. 3).

In late September 2017 the University of Potsdam installed a short period seismometer close to the Czech-German border in Oberzwota (red triangle, map 10) which is a favorable location. The station recorded 1000 samples per seconds for 62 days during a period of relative quiescence. Nevertheless, approximately 30 events were recorded in the swarm area with local

magnitudes down to Ml=0. Despite the installation directly on top of the weathering layer the recordings showed signal to noise ratios larger than 5 at 120 Hz and above for smallest magnitudes. It becomes evident that even a surface mounted station would allow to harness spectral information above the corner frequency of the WEBNET stations also for smallest magnitudes and which indicates that this will improve the resolution and robustness of our method once the ICDP borehole installations are operating.

**6   Conclusions**

Applying the source couple amplitude spectral ratio method to differential phase measures is an alternative to methods which commonly exploit the lower frequency ranges. Theoretically, it is therefore able to achieve better resolution. Our synthetic study validates this. The geometrical constraints of this method require a high density of events as it is the case for natural earthquake swarms or seismic nests but also for hydrofracturing experiments.

The application to data from the 2008 North West Bohemia earthquake swarm indicates source region $Q_p < 100$ based on measures at station *LBC*. This is in accordance to previous findings by Wcisło et al. (2018) who used station *NKC*. The sensitive region measures only approximately $2000 * 500 * 500$ meters in North, East and West direction (Fig. 2). Results can therefore be considered of high spatial resolution. Nevertheless, the distribution of solutions significantly scatter and we see room for

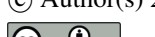

improvement e.g. through high frequency borehole recordings. We are not able to retrieve stable estimates at station *NKC* but instead see negative attenuation in southern and positive attenuation in the northern section of the swarm. *P* phase waveforms of the two sections show systematic differences at both significant stations which indicates a North-South structural difference. Furthermore, this effect does not inflict on measures at station *LBC*. Given the fact that ray segments at *NKC* and *LBC* probe

5 two different but directly neighboring media leads us to the conclusion that the fractured medium is highly concentrated along the source patch and that the surrounding medium can be considered much more dense or intact.

*Author contributions.* MK: Implementation, testing, evaluation and application to synthetic and recorded data, as well as paper writing. SC: Scientific supervision, evaluation of tests and applications. MO: Scientific supervision, discussion, manuscript revision. TD: Scientific supervision, discussion, manuscript revision. FK: Scientific supervision, discussion, manuscript revision.

10 *Competing interests.* None

*Acknowledgements.* This work is part of the HISS project which is funded by the DFG ICDP. Project no.: CE 223/2-1



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
