# Peer review of "Event couple spectral ratio Q method for earthquake clusters: application to North-West Bohemia"

_Solid Earth, 2018_

## Referee Comment (RC1) · Anonymous Referee #1 · 27 Oct 2018

Dear authors,

This paper demonstrates a new method to estimate attenuation (Q) using earthquakes. The authors exploit the dense hypocenter distribution of earthquake swarms in order to extract seismic wave attenuation at only a part of raypath at depth, which can achieve high resolution results. The authors demonstrate this approach using numerical modeling tests, and then apply it to field earthquake records at North-West Bohemia. The authors found that their results are very scattered, including the large number of negative Q values. They analyzed these results based on the location of events, and then found the correlation of negative Q values with the event location.

[Figure]

Estimation of the attenuation factor at high resolution is very important topic in applied seismology, and using seismic swarms to estimate local attenuation seems promising approach. The authors demonstrated the feasibility of the approach using well thought numerical modeling tests. Although the field application resulted in very scattered values in Q values, the authors discussed its reason considering several scenarios e.g., wave scattering and rupture dynamics. The authors' findings on the systematic changes of frequency contents depending on the event location would be interesting for various research groups, not limited to attenuation measurements.

Overall, the paper is reasonably well written. The processing techniques, figures and discussion are clear, and almost all conclusions are supported, although several discussions and explanations need to be more clarified. I have several comments, in order to further clarify technical details and the authors' claims. Please look at my comment below for more details.

Best regards.

Comments:

-Section 3 Synthetic study

1. Page 5, Line 23-25, "The size of blue points in Figure 3 represents relative number of pairs based on ray tracing through a 1D layered model": What the blue dots in Figure 3 indicate is not clear. Earthquake swarms are indicated by red dots. Please elaborate more on the definition of "event couples" in this figure.

2. Page 8, Line 7-8, "S phase results match the model at NKC but show strong scattering at LBC". Can you elaborate why this is the case?

3. Source time function is half sine shaped but the width depends on the magnitude, which seems realistic. Can you comment on how this source time function is representative in this field?

4. Velocity model is 1D (see Fig 4), and high frequency component that is used is

around 80 Hz (see Fig.9c). The short wavelength component is around 60 m assuming Vp=5000m/s in this case. In 1D velocity model used in the test does not include any spatial heterogeneity in the order of this short wavelength. Can you comment on to what extent we can ignore effects of spatial heterogeneity that is sensitive to the high frequency components in the data? I found that the authors discussed the effect of wave scattering in section 4 and 5. It would be good to mention here that these effects will be discussed later.

-Section 4 Application to North West Bohemia

5. Figure 9: I did not find sentences explaining Fig. 9 in the main text.

6. Page 11, lines 10-11, "It becomes evident that larger incidence angles (> 8 degrees) show a tendency to produce negative $Q^{-1}$ while results from events with steep incidence angles produce positive $Q^{-1}$ values." : Please mention that this refers to Figure 12(b), otherwise it does not make sense.

-Section 5 Interpretation and Discussion

7. Page 13, Lines 21-22, "Mousavi et al. (2017) assume a highly fractured medium in combination with accumulated free gas or fluids. Our findings support this hypothesis." : I did not see the clear relation between the authors results and this hypothesis. Please elaborate more.

Other minor comments:

-Page 5, Line 9 and 10: "(Dt, red dashed line, Fig 1)"-> (Dt, green dashed line, Fig 1), "(Dp, green dashed line, Fig 1)"-> (Dp, red dashed line, Fig 1)

-Page 8, Line 3, "Data windows without seismic events in the recorded data have been manually extracted...": I understand that the authors meant field recorded data by "the recorded data" in this sentence. I suggest explicitly mentioning so since it is confusing as to be synthetic data.

-Page 10, Line 10, "as Figure 11 (right) indicates" -> as Figure 11 (left) indicates?

---

## Referee Comment (RC2) · T. Fischer (Referee) · 31 Oct 2018

Review of manuscript
Event couple spectral ratio Q method for earthquake clusters: application to North-West Bohemia
by Kriegerowski et al., submitted to Solid Earth.

This is a novel study proposing and testing a new method for determining attenuation in the fault zone using spectral ratio of adjacent events, which works above the earthquake corner frequencies to remove dependence of spectra on the size of the events.
After synthetic tests the method is applied to the seismograms of the West Bohemia 2008 swarm. Rather large scatter of the resulting Q factors is found, which is discussed and attributed to possible scattering in the focal zone. It also appears that the method requires high quality data, so application to later seismicity in the area with denser recordings looks a promising way to improve the stability of the results.
Before accepting the study I recommend to better describe the method, the used data (space distribution of the hypocenters), the synthetic tests and improve the discussion. All unclear points are indicated below.

Questions/suggestions

- P3: the Method is described in a rather confusing way. Some examples:
  – $t^*$ in Eq(1) is different from $t^*$ in Eq(3). Instead $t^*_{ij}$ should be in Eq(1) and $t^* = t^*_{i0} - t^*_{i1}$ in Eq(3)
  – what do you mean by saying 'spectra S0 and S1 are similar' – I think they should be identical so that Eq(3) holds. To make it clear refer to 'below' where this is clarified
  - there is a mistake in Eq(3): logarithm is missing in front of $G_0/G_1$
- Fig. 2: the hypocenters are not visible: what about plotting them in the upper layer?
- P5/14: Please specify how are the upper frequency limits corrected to account for the Fresnel volume
- P5/23: The position of the blue points in Fig 3 is not clear; they do not appear to show the ray density or a similar parameter, because the rays cannot pass through these points. Please explain or correct.
- P7/3: Which three requirements do you have in mind? I have found only two in the preceding text
- P8: Please explain the difference between synthetic traces in Figs 5, 6 and 7. It is not clear which source time functions were used to generate Fig. 5, 6 and 7. Were these magnitude-dependent in Fig. 7 only?
  And what about Green functions? To my understanding, the synthetic traces in Fig.5 are computed using 1D model, which means (to my understanding) that the source time functions have been convolved with the appropriate Green function. So what is the difference in data for Figs 5 and 7?
  Possibly only two instead of three figures are necessary here?
- There is no reference in the text for Fig. 9
- Fig. 10: The swarm locations shown on the map look rather scattered – are these indeed the HypoDD locations you have used for the analysis?
- Fig. 11: How did you determine the incidence angles: from ray tracing or these are measured from the seismograms?
- P11/10: that larger -> that for larger
- P11/17: I think that the sentence "This trend is mostly dependent…" is not necessary
- Fig. 13: It would be more suitable to use a kilometer scale on this plot. Besides, the gap at 50.211 latitude appears enigmatic. I believe this plot should be station independent, because it shows coordinates of the events, so it should be visible at both the stations. Even if different events would be used at different stations, at least some indication of the gap should be visible also at

LBC, provided there is some overlap between the event sets. Could you please show the vertical section of used relocations in order to identify the origin of the gap?

- P12: I am not sure if only scattering is responsible for the reported waveform complexity at NKC. It could be caused by different effects. One of these could be the proximity to the nodal line, where the P-onsets are usually of emergent nature and later arrivals are more visible. The data shown however look quite impulsive, which could indicate that the two mechanisms have different nodal lines, which not close to the NKC projection. Another reason could be overlap of waveforms with opposite polarity, which makes the image more complex.

- P13/6: Here it would be helpful to discuss more the different ray direction and coverage of the focal zone for NKC and LBC as visible in Fig. 2, which could affect different sensitivity of the stations to attenuation.

- P14/1-8: It would be suitable to refer here to the study of Wcislo et al (2018) who obtained similar Qp and Qs using different method on the same data. Mentioning this in Conclusions is too late; BTW Conclusions usually do not contain citations.

Minor points
- Fig. 2 is referred only after Fig.4 on page 5
- P5/12 attenuation -> attenuated
- P10/10 (right) is superfluous

---

## Author Comment (AC1) · 14 Dec 2018

Dear Sir or Madam,

Thank you very much for your valuable comments. We addressed all your suggestions in the updated manuscript as well as below. Corrections in the manuscript are printed in red.

All references in the comments below refer to the references in the paper.

1. 1. Page 5, Line 23-25, "The size of blue points in Figure 3 represents relative number of pairs based on ray tracing through a 1D layered model": What the blue dots

in Figure 3 indicate is not clear. Earthquake swarms are indicated by red dots. Please elaborate more on the definition of "event couples" in this figure.

*We extended the text in this section as well as in the figure caption to make clear that the blue points' size scales with the number of arriving ray paths from event couples at theoretical station sites given the geometrical constraints described in the text.*

2. Page 8, Line 7-8, "S phase results match the model at NKC but show strong scattering at LBC". Can you elaborate why this is the case?

*We extended the underlying paragraph to clarify this: "[...] as a result of the interference with the added noise as well as the P phase coda".*

3. Source time function is half sine shaped but the width depends on the magnitude, which seems realistic. Can you comment on how this source time function is representative in this field?

*The source time durations we used in the synthetic modeling are based on estimates for the stress drop and source dimensions taken from the paper by Michálek (2013) for the 2000 - 2008 swarm activity and are therefore assumed to be representative for the given task.*

4. Velocity model is 1D (see Fig 4), and high frequency component that is used is around 80 Hz (see Fig.9c). The short wavelength component is around 60 m assuming Vp=5000m/s in this case. In 1D velocity model used in the test does not include any spatial heterogeneity in the order of this short wavelength. Can you comment on to what extent we can ignore effects of spatial heterogeneity that is sensitive to the high frequency components in the data? I found that the authors discussed the effect of wave scattering in section 4 and 5. It would be good to mention here that these effects will be discussed later

*We used the velocity model suggested by Alexandrakis (2014) which we assume is the most realistic 1D velocity model available. Small scale heterogeneities are expensive to model as these effects in the temporal and spectral domain can easily be outweight*

*by and also cause numerical instabilities. We added a comment on the following discussion in the text and also a sentence with regard to the actual waveforms recorded in NW Bohemia which is typified by clear onsets, as well (see Fischer (2010)).*

5. Figure 9: I did not find sentences explaining Fig. 9 in the main text.
*We added a sentence and a reference regarding that figure in the text.*

6. Page 11, lines 10-11, "It becomes evident that larger incidence angles (> 8 degrees) show a tendency to produce negative $Q\ddot{E}\xi{-1}$ while results from events with steep incidence angles produce positive $Q\ddot{E}\xi{-1}$ values." : Please mention that *this refers to Figure 12(b), otherwise it does not make sense. We extended that sentence to highlight that this refers to station NKC and explicitly refer to that figure.*

7. Page 13, Lines 21-22, "Mousavi et al. (2017) assume a highly fractured medium in combination with accumulated free gas or fluids. Our findings support this hypothesis." : I did not see the clear relation between the authors results and this hypothesis. Please elaborate more
*We clarified in the text that the hypothesis by Mousavi et al. (2017) would cause high source volume attenuation which is supported by our observations on P-wave attenuation.*

Other minor comments:
*All mentioned points have been corrected.*

Kind regards

Marius Kriegerowski and co-authors

---

## Author Comment (AC2) · 14 Dec 2018

Dear Mr. Fischer,

Thank you for valuable comments. We addressed all your suggestions in the text as well as below.

1. P3: the Method is described in a rather confusing way. Some examples: – $t^*$ in Eq(1) is different from $t^*$ in Eq(3). Instead $t^*_{ij}$ should be in Eq(1) and $t^* = t^*_{i0} – t^*_{i1}$ in Eq(3) – what do you mean by saying 'spectra S0 and S1 are similar' – I think they should be identical so that Eq(3) holds. To make it clear refer to 'below' where this is clarified -

there is a mistake in Eq(3): logarithm is missing in front of G0/G1
*We apologize and improved the methodological section considering all suggestions. We moved the clarification on what 'equal' means with respect to the two spectra to an earlier paragraph.*

2. Fig. 2: the hypocenters are not visible: what about plotting them in the upper layer?
*Done.*

3. P5/14: Please specify how are the upper frequency limits corrected to account for the Fresnel volume
*We have added one short paragraph to specify the role of the Fresnel volume to limit the frequency content.*

4. P5/23: The position of the blue points in Fig 3 is not clear; they do not appear to show the ray
*We rewrote the caption of figure 3 to clarify the meaning of the blue points.*

5. P7/3: Which three requirements do you have in mind? I have found only two in the preceding text
*Corrected.*

6. P8: Please explain the difference between synthetic traces in Figs 5, 6 and 7. It is not clear which source time functions were used to generate Fig. 5, 6 and 7. Were these magnitude-dependent in Fig. 7 only? And what about Green functions? To my understanding, the synthetic traces in Fig.5 are computed using 1D model, which means (to my understanding) that the source time functions have been convolved with the appropriate Green function. So what is the difference in data for Figs 5 and 7? Possibly only two instead of three figures are necessary here?
*We now explicitly mention in the caption and text the source time function used to plot figure 5, 6 and 7. This implies that figure 7 differs from the previous ones in the usage of the magnitude dependent source time function.*

7. There is no reference in the text for Fig. 9
*We added a reference and text introducing that figure.*

8. Fig. 10: The swarm locations shown on the map look rather scattered – are these indeed the HypoDD locations you have used for the analysis?
*The shown hypocentres are the hypoDD relocated events. We improved the map in general and use smaller points now.*

9. Fig. 11: How did you determine the incidence angles: from ray tracing or these are measured from the seismograms?
*We used a 1D raytracing algorithm. We clarified this in the text.*

10. P11/10: that larger -> that for larger
*corrected.*

11. P11/17: I think that the sentence "This trend is mostly dependent. . ." is not necessary
*We didn't remove this sentence as we think that this is a true feature. See reply no. 12.*

12. Fig. 13: It would be more suitable to use a kilometer scale on this plot. Besides, the gap at 50.211 latitude appears enigmatic. I believe this plot should be station independent, because it shows coordinates of the events, so it should be visible at both the stations. Even if different events would be used at different stations, at least some indication of the gap should be visible also at LBC, provided there is some overlap between the event sets. Could you please show the vertical section of used relocations in order to identify the origin of the gap?
*We use a kilometer scale now and modified the figure to show the contribution of deeper and shallower sources as a vertical section in the upper panels. The former latitude gap at station NKC was partially due to a plotting problem. However, some gap is still visible and resulting from the spatial non-homogeneous distribution of the deeper sources (Panel b)).*

13. P12: I am not sure if only scattering is responsible for the reported waveform complexity at NKC. It could be caused by different effects. One of these could be the proximity to the nodal line, where the P-onsets are usually of emergent nature and later arrivals are more visible. The data shown however look quite impulsive, which could indicate that the two mechanisms have different nodal lines, which not close to the NKC projection. Another reason could be overlap of waveforms with opposite polarity, which makes the image more complex.

*We actually consider different hypothesis to explain the complex signals at station NKC including scattering (P 13/13) but also focal mechanisms (P 13/5 ) and nodal lines (P 13/9). (Line numbers refer to the updated manuscript).*

14. • P13/6: Here it would be helpful to discuss more the different ray direction and coverage of the focal zone for NKC and LBC as visible in Fig. 2, which could affect different sensitivity of the stations to attenuation.

*We extended the paragraph discussing the focal zone penetration with respect to Figure 2.*

15. P14/1-8: It would be suitable to refer here to the study of Wcislo et al (2018) who obtained similar Qp and Qs using different method on the same data. Mentioning this in Conclusions is too late; BTW Conclusions usually do not contain citations.

*We agree and now cite the work of Wcislo earlier in the discussion.*

Best regards

Marius Kriegerowski and co-authors